Modeling the ecologic niche of plague in sylvan and domestic animal hosts to delineate sources of human exposure in the western United States

Walsh Michael 1 thegowda@gmail.com michael.walsh@downstate.edu
Haseeb MA 1 2
1 Department of Epidemiology and Biostatistics, School of Public Health, State University of New York,Downstate , United States
2 Departments of Cell Biology, Pathology and Medicine, College of Medicine, State University of New York, Downstate Medical Center , Brooklyn, NY , United States
Noymer Andrew
Electronic publication date: 2015 Dec 14
Publication date: 2015
Volume: 3
Electronic Location ID: e1493
Received 2015 Sep 12; Accepted 2015 Nov 20
Copyright: ©2015 Walsh and Haseeb
Copyright year: 2015
Copyright holder: Walsh and Haseeb
License: This is an open access article distributed under the terms of the Creative Commons Attribution License, which permits unrestricted use, distribution, reproduction and adaptation in any medium and for any purpose provided that it is properly attributed. For attribution, the original author(s), title, publication source (PeerJ) and either DOI or URL of the article must be cited.
License URL: https://creativecommons.org/licenses/by/4.0/

Keywords: Plague, Yersinia pestis, Landscape epidemiology, Zoonosis, Infection ecology, Peromyscus maniculatus

Funding: The authors received no funding for this work.

==============================
Plague has been established in the western United States (US) since 1900 following the West Coast introduction of commensal rodents infected with Yersinia pestis via early industrial shipping. Over the last century, plague ecology has transitioned through cycles of widespread human transmission, urban domestic transmission among commensal rodents, and ultimately settled into the predominantly sylvan foci that remain today where it is maintained alternatively by enzootic and epizootic transmission. While zoonotic transmission to humans is much less common in modern times, significant plague risk remains in parts of the western US. Moreover, risk to some threatened species that are part of the epizootic cycle can be quite substantive. This investigation attempted to predict the risk of plague across the western US by modeling the ecologic niche of plague in sylvan and domestic animals identified between 2000 and 2015. A Maxent machine learning algorithm was used to predict this niche based on climate, altitude, land cover, and the presence of an important enzootic species, Peromyscus maniculatus. This model demonstrated good predictive ability (AUC = 86%) and identified areas of high risk in central Colorado, north-central New Mexico, and southwestern and northeastern California. The presence of P. maniculatus, altitude, precipitation during the driest and wettest quarters, and distance to artificial surfaces, all contributed substantively to maximizing the gain function. These findings add to the known landscape epidemiology and infection ecology of plague in the western US and may suggest locations of particular risk to be targeted for wild and domestic animal intervention.

Introduction

Plague was introduced to North America in 1900 during the third major pandemic, which began in the 19th century in China (Eskey & Haas, 1940). Subsequently, the causative agent, Yersinia pestis, established in several foci across the semiarid regions of the western United States (US) where it is now maintained enzootically in large part by the deer mouse (Peromyscus maniculatus), with occasional epizootic events in some key species such as prairie dogs (Cynomys spp) and ground squirrels (Spermophilus spp.) (Eskey & Haas, 1940; Gage & Kosoy, 2005). Indeed, the geographic extent of the western US delineates one of the largest concentrations of sylvan plague foci in the modern world. Despite the extensive foci of Y. pestis in the US, there are far fewer zoonotic transmissions to humans than occur in other geographic areas of significant plague foci, such as Madagascar or the Democratic Republic of the Congo (Stenseth et al., 2008). Nevertheless, the median annual incident cases of plague in the US was three (range 0–17) between 2001 and 2012 (Kwit et al., 2015; Adams et al., 2014). Most of these human cases present as bubonic and can be quite severe, especially when not recognized and treated promptly with appropriate antibiotic therapy. As such, the ability to quantify the risk of zoonotic transmission from animals to humans across the region of established plague foci in the US remains a current research imperative. Flea bite and contact with infected animals accounted for 39.5% and 60.1%, respectively, of all infection transmissions with known exposure history between 1965 and 2012 (Kugeler et al., 2015). Moreover, while direct contact with wildlife is a recognized risk factor, plague is also often transmitted to domestic pets following exposure to wildlife, which then pass on the infection to their human caretakers (Craven et al., 1993). The importance of direct and indirect (via domestic animals) contact with wildlife notwithstanding, locations of animal plague in the landscape, which may act as conduits to zoonotic transmission of Y. pestis to humans, remain uncertain. Abiotic factors such as climate conditions and land cover and land use are of particular interest in understanding the distribution of animal plague. For example, relationships between animal plague occurrence and climate have been described in the western US, but in areas of more limited geographic extent (Holt et al., 2009; Eisen et al., 2007b; Eisen et al., 2007c), while studies of land cover have focused on sylvan host habitat without also considering proximity to developed land parcels (Eisen et al., 2007a). Equally important are biotic factors, such as the relative contribution of enzootic species (e.g., P. maniculatus) to plague occurrence among epizootic species (e.g., Cynomys spp.) (Stenseth et al., 2008; Gage & Kosoy, 2005). Therefore, it would be useful to attempt to model the presence of animal plague as a function of both abiotic and biotic features across the region. The current investigation sought to predict the ecologic niche of epizootic plague based on the presence of plague in sylvan and domestic animal hosts across all states west of the Mississippi River, and thereby identify spaces of potential zoonotic plague transmission to humans. A machine learning approach was used to model the ecologic niche as a function of climate, altitude, land cover, and the presence of the enzootic reservoir, P. maniculatus.

Materials and Methods

Sixty-six reports of laboratory confirmed, geolocated infection with Y. pestis in sylvan and domestic animal hosts were collected by the International Society of Infectious Diseases through the ProMED electronic surveillance system between January 1, 2000 and August 31, 2015 (http://www.promedmail.org/). Geographic coordinates for each case were obtained using Google Maps. These data, including the ProMED archive number for each report, are available via Figshare (http://dx.doi.org/10.6084/m9.figshare.1538604). For inclusion in this study, the animal cases were required to have been reported with a spatial resolution of 1 square kilometer or finer, meaning that cases reported at >1 km2 were of too coarse a resolution for consideration. In addition to the infections in animal hosts, there were 48 laboratory-confirmed plague cases in humans over the same period. However, these reports are generally only reported at the county level to protect patient privacy. Moreover, as many western US counties are large in area, the spatial resolution available for analysis using these human cases would be quite coarse, so human cases were not included in these analyses.

The WorldClim—Global Climate (2014) database was the source of all climate date used in this investigation. Annual mean temperature, annual mean precipitation, mean temperature for the hottest and coldest quarters, mean precipitation for the wettest and driest quarters, and isothermality from 1950 to 2000 were each extracted as 30 arc second resolution rasters (Hijmans et al., 2005). Altitude was also extracted from this database at the same resolution. Each pixel in these rasters represents the value of the measurement for that approximately 1 km2 area on the Earth’s surface.

A raster of the distribution of 9 unique land cover types was obtained from the Food and Agriculture Organization of the United Nations’ Global Land Cover—SHARE database (Latham et al., 2014). The resolution of this raster was 1 km2. The land cover types represented in this raster were as follows: bare soil, cropland, grassland, shrubland, sparse landscape, snow cover, tree cover, water, and artificial surface. Each of the 9 land cover types was extracted from this raster and the distance between each pixel and the nearest pixel of each unique land cover type was calculated to create a new distance raster, of the same extent and resolution, for each land cover type. The distance calculations were performed in the QGIS geographic information system (http://www.qgis.org/) using the proximity function for calculating raster distances. The result of this process was a separate raster of 1 km2 resolution for each of the 9 land cover types described above wherein the value of each pixel is the distance in meters between that pixel and the nearest pixel wherein the particular land cover feature is present. This allows for modelling a spectrum of proximity to different land cover types in a given space rather than the more crude approach of simply designating the space as present or absent for specific land cover (see modelling description below).

The Global Biodiversity Information Facility (GBIF) was used to identify the geographic distribution of one of the most important reservoir species of Y. pestis in the enzootic transmission cycle, Peromyscus maniculatus (http://www.gbif.org/). This database contained 94,983 field records of documented and geolocated P. maniculatus individuals within the spatial extent of latitude 49.38436°N, 25.83738°N, and longitude 88.81702°W, 124.7631°W. This extent delineates the boundaries of the western US within which all previous plague occurrences, both animal and human, have been documented. This field sample of 94,983 individual deer mice was used to create a raster of the predicted occurrence of the reservoir across the western US using a Maxent model (see description of statistical methods below).

Statistical analysis

A maximum entropy machine learning approach was employed to classify and map the ecologic niche of plague infection in sylvan and domestic animal hosts. Because the aim of this exercise was to predict the probability of animal plague occurrence across geographic space, machine learning is an attractive analytic approach. Machine learning does not require the assumptions of a specific model form. Algorithms are constructed to form decision trees, which partition a data space based on rules that optimally identify homogeneity among predictors and a response (i.e., the presence of plague infected animals) (Elith, Leathwick Hastie, 2008). The analytic structure of these decisions trees is appealing because they are robust to outliers, predictors may be of any form, and their hierarchical structure inherently models interactions between the predictors. The Maxent machine learning algorithm, as implemented in the dismo package in R (Hijmans et al., 2014), was used to predict plague in animal hosts in the western United States. More specifically, the Maxent algorithm was used to predict the probability that the biotic and abiotic conditions in a given location were suitable for the presence of animal plague. This algorithm is based on a maximum entropy probability distribution, which holds that in the absence of boundaries to species’ dispersal, the distribution of those species will approach uniformity. This algorithm is particularly appropriate for presence-only animal plague data because it does not require the unknown information associated with animal plague absences, which were not available (Phillips, Anderson & Schapire, 2006; Franklin, 2010). Documented species occurrences are interpreted as presence points, while those points without documented occurrences are treated as the background environment. In the Maxent model employed here, presence points are represented by the locations of identified animal plague cases. As described above, the geographic extent of this analysis was latitude 49.38436°N, 25.83738°N, and longitude 88.81702°W, 124.7631°W. Maxent models consistently compare favorably with other approaches to machine learning, such as random forests or support vector machines (Duan et al., 2014). The current analysis also compared the performance of Maxent to predictive models derived from random forests and support vector machines to determine if superior performance could be replicated in our study of animal plague.

For each 1 km2 geographic area within the spatial extent of the western US, the Maxent model was used (1) to predict the probability of habitat suitability of P. maniculatus, and (2) to predict the probability of epizootic plague. The first model, targeting P. maniculatus, included mean annual temperature, mean temperature during the coldest quarter, isothermality, mean precipitation during the driest quarter, mean precipitation during the wettest quarter, distances to water, grassland, shrubland, trees, and artificial surface as predictors of habitat suitability. The regularization parameter was set to 1.0, to balance between overly localized, overfit model predictions and overly generalized, broadly fit model predictions. This output raster described the ecologic niche of P. maniculatus at a resolution of 1 km2. Five-fold cross-validation was applied to the Maxent model, wherein the model was fit by first dividing the training sets into k = 5 subsets, and then cross-validating by iteratively fitting the model to 4 of the combined subsets and testing against a 5th. This was repeated such that each of the k = 5 subsets was used as a test set during one iteration. The final cross-validated model was then evaluated against a test dataset. The 5 subsets were comprised of 12,000 deer mouse observations each, with these being randomly selected from the overall pool of training observations (n = 60,000, or ∼2/3 of the total 94,983 deer mouse observations; see further description below). As a further evaluation of prediction error, the data were split into two subsets prior to beginning the analyses. The first subset was a training set (n = 60,000), which was used to fit (or “train”) the models, while the second subset (n = 34,983) was left out of the model fitting process altogether and then used to test against the model predictions. The difference in model predictions based on training and testing data provided an assessment of the model’s prediction error. The use of separate training and testing datasets reduces the typically high prediction error that attends overfitting of the data when all available data are used to train the model. In this analysis, the data were partitioned into 2/3 and 1/3 of the total observations for the training and test sets, respectively, and the area under the curve (AUC), reported as a percentage, was calculated to assess prediction error. The relative contribution is also reported for each predictor in the Maxent model. The relative contribution refers to the reduction of a loss function, which is referred to as the gain and is analogous to the residual deviance from generalized linear models, attributable to each predictor variable. The output raster of the predicted probability of P. maniculatus presence was then used as a predictor of epizootic plague. This subsequent Maxent model of epizootic plague included mean annual temperature, mean temperature during the coldest quarter, mean precipitation during the driest quarter, mean precipitation during the wettest quarter, distances to grassland, shrubland, cropland, sparse vegetation, bare soil, and artificial surface, and the predicted probability of P. maniculatus as predictors. The same 5-fold cross validation process used to model P. maniculatus probability described above was used to model epizootic plague. The animal plague data (n = 66) were also partitioned into a training set comprising 2/3 of the whole (n = 44) and a testing set comprising 1/3 of the whole (n = 22). As above, the AUC was reported to assess the prediction error for the epizootic plague model and the relative contribution was reported for each predictor variable in the model. Variable response curves were also examined to more clearly elucidate specific relationships between individual variables and predicted animal plague occurrence.

Model predictions of epizootic plague probability were converted to a binary score to designate presence versus absence across a range of predicted plague occurrence. Four thresholds were thus created at 25%, 50%, 75%, and 90% predicted probability. These thresholds were used to create four separate binary rasters designating each 1 km2 space as either present or absent for epizootic plague.

Results

The distribution of the Y. pestis infected animals as reported by the ProMED electronic surveillance system of the ISID is presented in Fig. 1. As expected, the distribution is limited to the western United States, but with high concentrations of reported infected animals in Colorado, New Mexico, and southern California. Central South Dakota marked the eastern boundary of reported animal plague during the 15 year surveillance period, while north-central Montana delineated the northern boundary in close proximity to the Canadian border (Hijmans et al., 2005).

Figure 1 The distribution of the 66 laboratory-confirmed animal plague cases identified through the ProMED system between January 1, 2000 and August 31, 2015 in the United States.

Map data: Google, ©2015 TerraMetrics.

Figure 2 Rasters of model predictors.

The predicted distribution of Peromyscus maniculatus (deer mouse), altitude, mean precipitation during the driest and wettest quarters, distances to land cover types, and mean annual temperature with the distribution of the 66 animal plague cases overlaid (dots).

Maps of the distribution of the environmental variables with the locations of infected animals overlaid are presented in Fig. 2. Some of the environmental features clearly demonstrate greater heterogeneity than others. For example, precipitation, in both the driest and wettest quarters, and temperature were unevenly distributed across the western states, as was proximity to artificial surface, grassland, and bare soil land cover types. The map of predicted P. maniculatus in the first panel depicts the probability surface of this enzootic species across the western US. The Maxent model used to predict the species distribution identified mean annual temperature, isothermality, and distances to artificial surfaces, grassland, and shrubland as the most influential with 34.7%, 22.6%, 14.9%,9%, and 8.9% relative contribution to the loss function, respectively. The AUC for the P. maniculatus species distribution model was 76% indicating acceptable prediction error against test data.

Figure 3 Predicted probability of epizootic animal plague.

These risk surfaces are based on the ecologic niche of animal plague as derived from the Maxent model.

The ecologic niche of epizootic plague as predicted by the Maxent model is presented in Figs. 3 and 4. The former delineates a continuous risk surface as the probability of animal plague occurrence, while the latter figure depicts animal plague presence versus absence based on four thresholds of 25%, 50%, 75%, and 90% predicted probability of animal plague occurrence. In all maps counties are included to provide a frame of reference within specific local municipalities. These maps highlight a pattern of occurrence, albeit somewhat discontinuous, predicted along the eastern edge of the front range of the American Rocky Mountains that extends from northwestern Montana down to north-central New Mexico. Additional areas of high occurrence include southwestern and northeastern California, southwestern South Dakota, central Arizona, as well as more localized areas in Oregon, Washington, southern Idaho, and northern Utah. As the threshold for designating animal plague presence increased from 25% to 90%, the range of predicted presence narrowed until it was limited primarily to the front range within the state of Colorado, with a few additional highly localized areas in Oregon, California, and New Mexico. In the Maxent model used to predict the ecologic niche described in these maps, predicted presence of P.maniculatus, altitude, distance to artificial surfaces, mean precipitation during the wettest quarter, and mean precipitation during the driest quarter were the most influential environmental variables contributing 50%, 17.2%, 8.3%, 7.2%, and 7%, respectively, to the loss function. Additionally, mean annual temperature and distances to cropland, bare soil, and sparse vegetation modestly contributed between 2%–3%, while temperature during the coldest quarter and distances to grassland and shrubland all contributed less than 1%. The AUC derived from this Maxent model was86%, with confidence bands based on k-fold cross-validation replicates between 85.9% and 86.1%, indicating acceptable prediction error against the test data. Moreover, the Maxent model performed better than either a random forests model (AUC = 83%) or a support vector machine model (AUC = 70%).

Figure 4 Predicted presence of animal plague based on thresholds of 25%, 50%, 75%, and 90% probability of animal plague occurrence derived from the Maxent model.

The variable response curves highlighted some interesting landscape features of habitat suitability for animal plague (Fig. S1). The relationship between altitude and predicted animal plague probability demonstrated a threshold at approximately 2,000 m, below which plague risk increased linearly from sea level and after which no further increase in risk was observed. Thresholds were also noted for precipitation. Animal plague probability increased modestly with increasing mean precipitation during the driest quarter up to 50 mm, while falling precipitously after this threshold until reaching 0% probability of plague by 100 mm. Conversely, mean precipitation during the wettest quarter was associated with a sharp increase in predicted probability from 0 mm to 100 mm of precipitation, while steadily falling back down to zero between 100 mm and 1,200 mm of precipitation. The sensitivity analysis, presented in Fig. S2, shows the probability of predicted plague occurrence at the 5 km2 predictor resolution to be very similar to that of the 1 km2 predictor resolution, including the same abiotic and biotic predictors in both models. Moreover, the AUC for this lower resolution model was 88%. Therefore, this model of plague appears to be robust to scale.

Discussion

This investigation mapped the ecologic niche of epizootic plague across the western United States based on the presence of Y. pestis infection in sylvan and domestic animal hosts. The presence of P. maniculatus, mean precipitation during the driest and wettest quarters, altitude, and distance to artificial surfaces delineated influential landscapes with respect to the prediction of animal plague occurrence.

Plague foci in the western US have been recognized in sylvan species for several decades (Eskey & Haas, 1940; Eisen et al., 2007a). The transmission cycle of Y. pestis is defined by multi-directional, complex pathways that involve enzootic and epizootic transmission in animal hosts, and zoonotic and person-to-person transmission in humans (Stenseth et al., 2008; Gage & Kosoy, 2005). The enzootic cycle involves vector-borne transmission of the pathogen between fleas and burrowing rodent species believed to be resistant to Y. pesits pathogenesis, particularly the deer mouse (P. maniculatus) and the California vole (Microtus californicus) (Gage & Kosoy, 2005; Poland, Quan & Barnes, 1994; Quan & Kartman, 1956; Quan & Kartman, 1962). The epizootic cycle emerges periodically in species highly susceptible to the pathogen and typically results in rapid and widespread transmission, and large die-offs. The species most affected by these epizootics in the western US are prairie dogs (Cynomys spp.), the black-footed ferret (Mustela nigripes), the California ground squirrel (Spermophilus beecheyi), and the rock squirrel (Spermophilus variegatus) (Gage & Kosoy, 2005; Quan et al., 1985; Poland, Quan & Barnes, 1994). The enzootic and epizootic cycles are both maintained locally through flea-borne transmission. However, more widespread diffusion beyond local sylvan populations is possible via transport of fleas by predatory or passerine birds or via sylvan mammal predators or domestic cats that prey on sylvan hosts (Stenseth et al., 2008; Gage & Kosoy, 2005; others). Zoonotic transmission of bubonic plague to humans is vector-borne or contact mediated. Vector-borne zoonotic transmission typically results from fleas acquiring infection from commensal rodents (e.g., Rattus norvegicus) and then passing the infection on to humans following the death of the commensal, while contact transmission requires contact with an infected animal, either domestic or sylvan (often by handling the dead carcass in the case of sylvan hosts) (Stenseth et al., 2008; Gage & Kosoy, 2005). Domestic cats are the primary source of domestic zoonoses of plague as they (1) often prey on wildlife hosts, and (2) are highly susceptible to Y. pestis infection and pathogenesis. Interestingly, pneumonic plague can also be transmitted from domestic cats to humans, but is more commonly transmitted person-to-person. These transmission cycles of plague ecology reveal several animal hosts that may be relevant for zoonotic transmission to humans, but the emergence of epizootics is generally believed to drive spillover to humans (Gage & Kosoy, 2005; Kugeler et al., 2015). As such, species involved with the epizootic transmission cycle were the current focus in attempting to identify areas of high animal plague occurrence in the landscape, which may have the potential drive plague spillover to humans. The current study examined plague in animals relevant to this epizootic cycle (squirrels, prairie dogs, and sylvan predators), or those domestic animals (primarily cats) that may acquire the infection via their intersection with the epizootic cycle and which may then pass on the infection to their human caretakers. Plague spillover to humans via the enzootic cycle is believed to be a rare occurrence and so the relevance of species within this cycle to the human risk surface is probably minimal (Gage & Kosoy, 2005). Nevertheless, species in this cycle may play a role in the force of infection for highly susceptible species in the epizootic cycle and, therefore, the ecologic niche of one such species (P. maniculatus) was used to predict the presence of epizootic plague. While we did not assess the force of infection directly in this study, our findings did show that P. maniculatus presence was the greatest contributor to the probability of animal plague occurrence in highly susceptible animals.

Altitude has been previously identified as an important feature of the landscape epidemiology of plague in the western US (Eisen et al., 2007c; Eisen et al., 2007a) and in other parts of the Americas (Schneider et al., 2014). While increasing elevation has been associated with a greater occurrence of plague, most investigations have identified an elevation ceiling, beyond which plague occurrence either no longer increases, or reverses and begins to decrease (Eisen et al., 2007a; Eisen et al., 2007c; Schneider et al., 2014). Indeed, a similar threshold was demonstrated in the current study. Increasing elevation was linearly associated with increasing probability of animal plague until a threshold was reached at 2,000 m, after which the probability of occurrence reached a plateau. The reason for such a threshold is not entirely clear, but may coincide with generally increasingly favorable habitat conditions with increasing altitude from sea level ultimately giving way subsequently to more limited distribution of relevant habitat and host species in areas of the highest altitudes. For example, certain tree species such as pinyon pines or juniper pines are preferred by P. maniculatus in southwestern landscapes, which occur within a range of moderate elevation and which may influence the occurrence of epizootic animal plague via the distribution of an important enzootic reservoir (Eisen et al., 2007b; Eisen et al., 2007a).

As with altitude, climate has been previously shown to be an important predictor of both sylvan plague and zoonotic transmission to humans (Eisen et al., 2007b; Enscore et al., 2002; Holt et al., 2009; Parmenter et al., 1999). These studies have generally found that increasing precipitation during the winter and spring months were associated with increasing incidence of plague in the following summer. In the current study, increasing mean precipitation during the wettest quarter was associated with increasing plague risk up to a threshold of 100 mm in a three month period. After this threshold, however, plague risk begins to steadily decline in a linear fashion with increasing mean precipitation during the wettest period. Increasing mean precipitation during the driest quarter also corresponded to a more moderate increase in plague risk up to a threshold of 50 mm, after which plague risk dropped very quickly back to zero. Similar precipitation patterns were observed previously in California (Holt et al., 2009) and New Mexico (Parmenter et al., 1999). This may suggest that the importance of precipitation to plague risk is seasonal with more rain during the wet period perhaps influencing population dynamics in sylvan host species and subsequent plague transmission cycles. In particular, periods of higher precipitation may correspond to subsequent periods of greater food abundance, which may increase the range and volume of important enzootic species or epizootic species. This would be expected to be particularly relevant for P. maniculatus in its favored pinyon-juniper woodlands in the southwest due to increased production of pinyon seeds. Parmenter et al. (1999) showed that increasing precipitation was an important precursor to emergent human infections, but not epizootics as such. Moreover, their findings only held at highly localized, large scales of analysis rather than across broad geographic context.

Proximity to artificial surfaces was associated with increasing animal plague occurrence. Moreover, this was the most influential land cover type (relative contribution = 8.3%), as compared with grassland, cropland, shrubland, and bare soil. To the best of the authors’ knowledge, this is the first study to demonstrate an influence of developed land on animal plague occurrence in the US. Landscape fragmentation has previously been shown to be associated with animal and human plague occurrence in other parts of the world with significant plague foci, such as Madagascar (Duplantier et al., 2005) and Tanzania (McCauley et al., 2015). Increasingly developed landscapes encroach on natural habitat and usher changes that may increase contact with sylvan species, which in turn may expand previously closed or limited transmission cycles and lead to zoonoses (Daszak, Cunningham & Hyatt, 2000; Estrada-Peña et al., 2014; Wolfe, Dunavan & Diamond, 2007; Woolhouse & Gowtage-sequeria, 2005). While the current investigation did not measure fragmentation directly, the distribution of developed land cover, as measured by artificial surfaces, may serve as a proxy for fragmentation and, thus, provide initial support for the role of human pressure in the infection ecology and landscape epidemiology of plague in the western US.

It is important to recognize several limitations inherent in the current study. First, the rasters for temperature and precipitation consisted of single composite measures over the period 1950 to 2000. Given that these rasters were averaged over a 50 year time span, the temporal resolution was coarse even though the spatial resolution was fine (∼1 km2). Nevertheless, the measures of temperature and precipitation in this study are believed to be sound assessments of the general climate of the western US, and thus provide a robust, if somewhat conservative and temporally coarse, approach to quantifying background climate. Second, this study relies on reports that are archived in the ProMED surveillance system and, thus, may have missed animal infections of Y. pestis not reported by local public health departments or by land and wildlife management authorities. Third, the number of occurrences of animal plague reported between 2000 and 2015 is relatively small, providing a sample size of only 66 precisely geolocated animal cases. Therefore, this collection of sylvan and domestic animal infections may not be representative of the total potential events in the western US. Nevertheless, this analysis was thorough in validating the sparse data by (1) using 5-fold cross-validation (2) validating the training models against test data, which showed low prediction error (AUC = 86%), and (3) analyzing the landscape variables at 2 different resolutions (1 km2 and 5 km2) to show consistency of the predictions across scale. Furthermore, the Maxent model is robust to spatial sampling bias that may be inherent in presence-only species distribution data. Most importantly, we do not purport these data to represent the definitive picture of animal plague presence in the western US. Rather, these findings are recognized as tentative but, nevertheless, useful to the evolving understanding of the landscape epidemiology and infection ecology of plague in sylvan and domestic animals in the western US. It is equally recognized that this model of plague risk will require the scrutiny of ongoing validation as more data become available. More specifically, direct measurement of animals in the specific locations of distinct enzootic and epizootic transmission cycles must be made, particularly with a view toward interactions between these cycles and subsequent zoonotic transmission to humans. This endeavor will require broad field sampling to record animal serology, habitat description at a much finer resolution and richer detail (i.e., describing not simply land cover, but providing more precise descriptions of habitat parcels in terms of biotic and abiotic properties), specific occupational and social practices, and residential circumstances, that bring humans into close proximity to animal hosts. Such research will be expensive, but it is necessary if we are to generate rich data suited to inform wildlife and domestic interventions that can limit dangerous human-animal encounters and block zoonotic plague transmission. Localized field investigations into climate and altitude have been conducted in the western US and generally support the findings from the current study, as described above. However, these results corroborate, but do not validate, the current findings. More extensive field studies conducted within the same geographic extent of the western US will be required to validate our predictions. In particular, areas designated to have a particularly high probability of animal plague occurrence (e.g.,75%–90%) may be targeted by local health departments and wildlife management authorities for animal trapping and serologic testing. The predicted high risk areas could then be compared to adjacent areas predicted to have a low probability of occurrence.

Conclusions

This investigation predicted animal plague occurrence across the western US based on reported occurrences of plague in sylvan and domestic animal hosts. The distribution of P. maniculatus, presumed to be an important species in the enzootic cycle of plague, altitude, precipitation during both the driest and wettest quarters, and proximity to developed land parcels were identified as important features of habitat suitability for animal plague, and may demarcate potential areas of zoonotic transmission to humans.

Supplemental Information

Figure S1 Variable response curves for altitude and precipitation

Click here for additional data file.

Figure S2 Predicted probability of epizootic animal plague at 5 km2 resolution

These risk surfaces are based on the ecologic niche of animal plague as derived from the Maxent model. All predictions are based on landscape variables aggregated to 5 km2 resolution.

Click here for additional data file.

Supplemental Information 1 Animal plague data

Click here for additional data file.

Additional Information and Declarations

Competing Interests

Author Contributions

Data Availability

The authors declare there are no competing interests.

Michael Walsh analyzed the data, wrote the paper, prepared figures and/or tables, reviewed drafts of the paper.

MA Haseeb wrote the paper, reviewed drafts of the paper.

The following information was supplied regarding data availability:

http://dx.doi.org/10.6084/m9.figshare.1538604.

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
