# Peer review of "Modeling the ecologic niche of plague in sylvan and domestic animal hosts to delineate sources of human exposure in the western United States"

_PeerJ, doi:10.7717/peerj.1493_

## Round 0.1 · original submission · Major Revisions

· Academic Editor

Major Revisions

You have received opinions of two qualified referees, one of whom has chosen to remain anonymous.

The reviewers make a number of good points, that can be used to revise and improve the paper. I will not re-enumerate all the concerns; you have the report. The questions about: whether the model can produce interval estimates; the ecological meaning of altitude at different locales; the interpretation of population density with county-level data, especially in the American west where counties are extremely heterogeneous in size and where population is almost a point source within some counties; and the interpretation of the AUC numbers, strike me as especially constructive comments.

Please address these and other issues raised by the reviewers. As editor, I can tell you that my philosophy regarding revisions is that all objections raised must be "addressed" but that this does not necessarily mean changing the paper on each point. It may be feasible in some cases to explain in your response memo (and/or revised paper), why a proposed change is not necessary, wise, or feasible. Such explanations must convince, however.

·

Basic reporting

The paper "Dilineating the ecological niche of sylvan and domestic animal plague to apprise human risk in the western United Sates" uses previous records of animal plague (the title makes this sound like two different plagues?) and modeling to document and map where plague is most likely to occur, and to link cases to ecological and socioeconomic variables that may be driving presence/prevalence of the disease. Although human cases are mentioned, they are not included in any part of the analyses, but authors still attempt to link models to human cases by overlaying the probability of occurrence derived from models on to a map of human population density.
Overall, there is some good analyses and ideas here, but I do not think the authors fully understand either the models or what the overall goal of their work might be. They use Maxent as their distribution model, but their use of the terminology to describe their results is shaky. Commenting that Maxent was used to predict "species density" or "the probability density of habitat suitability" suggests some lack of knowledge about what Maxent is actually doing, and the fact that the paper fails to cite Phillips et. al 2006 (the original author of Maxent) further supports this. It is very easy to include presence locations and build a species distribution map, but the authors must do a better job describing what is actually happening here so that readers can understand the advantages and limitations of this particular method.
Given the fact that the paper uses existing data and a modeling method that has been extensively used, the authors must do a better job of describing how this work is novel or appealing; what is it, as readers, that the authors would like us to learn from this work? As it stands, the paper reads more like a tutorial on how to do SDM without much in the way of driving hypotheses or revealing results.

Experimental design

The experimental design is not flawed, but I would be curious to know what resolution the animal cases had in terms of where the animal was recovered. Authors mention they got data from proMed, but as far as I know these records are only at the county level, which makes analyses difficult. Did authors get the exact locations of the cadavers for this analysis? Otherwise, linking those cases to 1-km spatial resolution can be tough. Please be explicit as to how precise geolocation data was. In general, Methods could be much clearer (for example I am not sure what's being said in Lines 155-158). Rather than suggesting Maxent is the best model to use, why not either formally or informally test other models to see if any striking differences are found (this has been done formally in Elith et al. 2011 and Harrigan et al. 2014 for other systems).

Results are presented at a somewhat superficial level, for instance, the thresholds mentioned could be more thoroughly investigated just by examining the variable importance curves and variable output from Maxent models. Why suggest this could be a possibility and then not see what the models tell you about this? Similarly, precipitation is treated somewhat vaguely and there is no interpretation of how plague itself or hosts might be affected by precip., rather than just a reference that suggests precip. can be important...

Validity of the findings

The validity of findings are likely there, but the interpretation needs careful consideration. Results are presented in a draft-like format, with paragraphs broken down by just Figure descriptions one after another (Figure 1 shows....Figure 2 presents....etc.). Authors should do a better job integrating Results, and again, focusing the reader on the most salient results.

A cutoff of 50% was used to say whether a pixel should be labeled as "plague" or not. The cutoff is not justified nor normalized in any way, and it is likely that the probability of occurrence did not go from 0 to 100, so this value does not even represent a mid-point. Why was this chosen?

Why are results and models listed as "good" or "very good" because they have a certain AUC score? There is no standard for these types of models, and no comparison was made within this data, or using other data, to justify these statement. For instance, authors could determine how little data is required to get a "good" model, perhaps they would be surprised at how easy, or difficult, it is to get similar performing models.

Authors find a 34% contribution of temperature, as well as precipitation, to predicting probability of plague occurrence. Why not take this to the next logical step and use this relationship to predict where we might see plague under future climate conditions? You have the model, the IPCC 5th assessment is publicly available, to me this is low-hanging fruit that could have a dramatic increase in the interest in this work, and could serve as a nice working hypothesis (see above).

The connection to human plague and the overlay on human demography is, to me, not an appropriate way to analyze this. As the authors themselves state, population density at the county level is highly-biased when presented as a single color. In many western counties, high population density is concentrated in one or two cities (think Nevada, it's Las Vegas and Reno) with little population outside those cities. Risk to humans should be done more appropriately than this; using some kind of kernel density method or actual human cases, but not with just an overlay. In fact, my recommendation is to abandon this connection altogether and instead focus on analyses you can actually tie in to your modeling efforts (the climate change work above would be a way to do this).

Discussion lacks any interpretation across much of the writing. For instance, Lines 272-308 reads more like background or introductory material, there is no mention of the actual results here.

Comments for the author

Overall, the paper presents a solid backbone, but the writing and interpretation should be thoughtfully considered. I do not mean to sound overly harsh; I think we these improvements, the paper will be of much higher caliber, and certainly more frequently cited.

Reviewer 2 ·

Basic reporting

In general, this was a well-written manuscript that provided a clear overview of the rationale and research design, which in its details necessarily would only be accessible for a specialist.
Some the discussion repeated information that was in the introduction, and the legends for the last two figures should be more precise about what "predicted plague" means. It might be helpful for readers if the authors restricted use of "plague" to the human disease and used "Y. pestis infection" for enzootic and epizootic infections.

Experimental design

Wherever possible confidence intervals for model estimates should be provided.
Unless I missed it, I did not understand why the authors did not examine the interaction of altitude and latitude. 2000 meters in San Diego County in California is a very different environment from 2000 meters in the northern Rockies.

Validity of the findings

The authors discuss limitations of the study and provide a general outline for how finer grained estimates could be obtained. But it would seem that with a modest effort there could be a limited seroprevalence study carried out in an area predicted to be suitable for Y. pestis transmission, but not as yet documented (e.g. northeastern California) and in an adjoining control area predicted by the model to be of low risk. State health departments commonly carry out trapping and blood sampling to detect the presence of Y. pestis in wildlife. Of course, this need not be done for this piece of work, but I think the authors should point out a feasible way for their model's predictions to be falsified.

---

## Round 0.2 · Minor Revisions

· Academic Editor

Minor Revisions

I am returning this manuscript to you with MINOR REVISIONS. Informally, you can consider this a "conditional accept", which is a category that some journals have, which PeerJ does not (officially) have.

Both referees are pleased with your changes, as am I. One of the referees requests a clarification on one point. I agree that it merits clarification.

Please revise the manuscript to address this one point. The revised and resubmitted manuscript may receive further limited peer review.

Please note that as an editor, my philosophy is NOT to go through endless rounds of reviews. I can assure you that I do not intend to send this out to new referees who will then find new objections, and so on. This sometimes happens and I know first-hand that it is one of the most frustrating things that can occur to a scientist.

It just seems to me that the referee's one remaining reservation is clearly stated and that you deserve the chance to clarify.

Thank you very much and I look forward to receiving your revised manuscript.

·

Basic reporting

Authors have done an admirable job of addressing the major issues that I previously had, they are to be commended on the edits. I think excluding the human overlays allows the work to stand better on its own and avoid conjecture.

Experimental design

Still quite confused on the animal case reporting (Lines 120-122). A "minimum" spatial scale of 1 square kilometer implies that cases could have had a 2km, 3km, or 10km squared spatial resolution and still be included. But then the very next sentence it is said that cases with >1km are "too small in scale" and too course a resolution to be considered. This is ultimately very confusing and is impossible to interpret.

I think authors mean to say that only cases that had a 1 km2 spatial resolution or finer were considered, and any records with coarser spatial resolution (>1km2) were excluded. If not, then please correct this. But a simple sentence like this is required.

Validity of the findings

As stated above, all validity of findings are much more transparent now with the exclusion of human overlays. Much better job in interpretation of AUCs and reporting.

Comments for the author

Again, the authors are to be commended for a much more thorough interpretation and reporting of findings. I hope the authors appreciate how improved the manuscript is, and I only have the one clarification above as an additional comment.

Reviewer 2 ·

Basic reporting

The concerns of my review have been either addressed in the response and with changes in the manuscript. And while not the expert on the methodology that reviewer #1 is, it seemed to me that the authors suitably responded to that reviewer's critique.

Experimental design

Please see comment above.

Validity of the findings

Please see comment above.

Comments for the author

The concerns of my review have been either addressed in the response and with changes in the manuscript.

---

## Round 0.3 · accepted · Accept

· Academic Editor

Accept

Thank you for your diligent attention to the reviewer comments.

·

Basic reporting

Good

Experimental design

Wording makes methods much easier to understand now. Thanks.

Validity of the findings

Good

Comments for the author

Thanks for the changes much easier to read and understand now.